# A Secondary Retrospective Analysis of the Predictive Value of Neutrophil-Reactive Intensity (NEUT-RI) in Septic and Non-Septic Patients in Intensive Care

**DOI:** 10.3390/diagnostics14080821

**Published:** 2024-04-16

**Authors:** Paolo Formenti, Letizia Isidori, Stefano Pastori, Vincenzo Roccaforte, Elena Alessandra Mantovani, Massimiliano Iezzi, Alessandro Menozzi, Rossella Panella, Andrea Galimberti, Giovanni Brenna, Michele Umbrello, Angelo Pezzi, Francesco Vetrone, Giovanni Sabbatini, Miriam Gotti

**Affiliations:** 1S.C. Anestesia, Rianimazione e Terapia Intensiva, ASST Nord Milano, Ospedale Bassini, 20097 Cinisello Balsamo, Italy; letizia.isidori@asst-nordmilano.it (L.I.); andrea.galimberti@asst-nordmilano.it (A.G.);; 2S.C. Analisi Chimico Cliniche e Microbiologiche, ASST Nord Milano, Ospedale Bassini, 20097 Cinisello Balsamo, Italy; stefano.pastori@asst-nordmilano.it (S.P.);; 3School of Medicine and Surgery, University of Milano-Bicocca, 20126 Milano, Italya.menozzi2@campus.unimib.it (A.M.); 4Department of Intensive Care, New Hospital of Legnano (Ospedale Nuovo di Legnano), 20025 Legnano, Italy; michele.umbrello@fastwebnet.it

**Keywords:** sepsis, neutrophil reactivity (NEUT-RI), C-reactive protein, procalcitonin

## Abstract

Background: Effective identification and management in the early stages of sepsis are critical for achieving positive outcomes. In this context, neutrophil-reactive intensity (NEUT-RI) emerges as a promising and easily interpretable parameter. This study aimed to assess the predictive value of NEUT-RI in diagnosing sepsis and to evaluate its prognostic significance in distinguishing 28-day mortality outcomes. Materials: This study is a secondary, retrospective, observational analysis. Clinical data upon ICU admission were collected. We enrolled septic patients and a control group of critically ill patients without sepsis criteria. The patients were divided into subgroups based on renal function for biomarker evaluation with 28-day outcomes reported for septic and non-septic patients. Results: A total of 200 patients were included in this study. A significant difference between the “septic” and “non-septic” groups was detected in the NEUT-RI plasma concentration (53.80 [49.65–59.05] vs. 48.00 [46.00–49.90] FI, *p* < 0.001, respectively). NEUT-RI and procalcitonin (PCT) distinguished between not complicated sepsis and septic shock (PCT 1.71 [0.42–12.09] vs. 32.59 [8.83–100.00], <0.001 and NEUT-RI 51.50 [47.80–56.30] vs. 56.20 [52.30–61.92], *p* = 0.005). NEUT-RI, PCT, and CRP values were significantly different in patients with “renal failure”. NEUT-RI and PCT at admission in the ICU in the septic group were higher in patients who died (58.80 [53.85–73.10] vs. 53.05 [48.90–57.22], *p* = 0.005 and 39.56 [17.39–83.72] vs. 3.22 [0.59–32.32], *p* = 0.002, respectively). Both NEUT-RI and PCT showed a high negative predictive value and low positive predictive value. Conclusions: The inflammatory biomarkers assessed in this study offer valuable support in the early diagnosis of sepsis and could have a possible role in anticipating the outcome. NEUT-RI elevation appears particularly promising for early sepsis detection and severity discrimination upon admission.

## 1. Introduction

Sepsis represents a severe medical condition marked by significant organ dysfunction and a potentially life-threatening state, arising from an uncontrolled host reaction to infection [1]. Sepsis and septic shock pose significant global health challenges, leading to mortality in one-third to one-sixth of affected individuals [2]. A Sequential Organ Failure Assessment (SOFA) score of ≥2 indicates a considerable likelihood of mortality and a hospital admission rate surpassing 10% according to the revised sepsis definition [3]. Timely identification and appropriate management during the initial stages of sepsis development are pivotal for favorable outcomes [4]. However, an accurate diagnosis is crucial to prevent unwarranted antibiotic use and, therefore, the outbreak of antibiotic resistance [5]. Currently, blood cultures are considered the gold standard for pathogen isolation and sepsis diagnosis. Although often started empirically, microbiologic cultures are crucial for adjusting the choice of antibiotic therapy [6]. Routine sepsis markers, such as white blood cell count (WBC) and C-reactive protein (CRP), lack specificity [7]. CRP, commonly used as a biomarker for acute inflammatory states, exhibits increased plasma concentrations parallel to the infection’s clinical course with a decrease indicating resolution [8]. However, its specificity is limited, necessitating the evaluation of alternative markers, like procalcitonin (PCT), tumor necrosis factor (TNF), and interleukin 6 (IL-6), although with challenges like cost and processing time [9,10,11,12,13]. Moreover, PCT and CRP levels may be influenced by renal insufficiency. Recent research has shifted focus to the role of neutrophil granulocytes in inflammation [14,15], particularly the neutrophil-reactive intensity (NEUT-RI) [16]. NEUT-RI emerges as a promising and easily interpretable parameter with literature supporting its correlation with an increase in immunoglobulins [17]. In fact, NEUT-RI reflects the metabolic activity of a neutrophil population by measuring the fluorescence intensity (FI). Notably, NEUT-RI exhibits a significant correlation with infectious status [18], strengthening its potential utility in early sepsis diagnosis and facilitating prompt initiation of optimal therapeutic interventions. The primary objective of this study was to evaluate the predictive value of NEUT-RI in sepsis diagnosis compared to other commonly used inflammatory parameters based on our previous observation [19]. Additionally, the secondary aim was assessing the prognostic value of NEUT-RI in discriminating 28-day mortality outcomes after intensive care unit (ICU) admission.

## 2. Materials and Methods

A comprehensive, retrospective, observational analysis of electronic medical records was performed to evaluate the diagnostic efficacy of NEUT-RI. The existing database, including patient information spanning from March 2022 to November 2022, underwent thorough scrutiny and augmentation with data collection extending until October 2023 [19]. The clinical data of patients upon admission to the ICU (clinical diagnosis, proximal and remote medical history for enrollment and exclusion criteria from the study), their respective laboratory values (WBC, CRP, PCT, extended inflammatory parameters, creatinine), blood culture results, SOFA score, and SAPS II for severity assessment were collected. NEUT-RI and PCT were also recorded at 48 h and 96 h from admission. The infection site and the 28-day outcome were added to this existing data. Data were collected retrospectively and stored in a dedicated database for consecutively admitted patients to two ICUs in the Milan area (ASST Nord Milano). The protocol for this study was examined and approved by the Ethics Committee of Milano Area 3 (n.114/2023, 7 February 2023). We included consecutive ICU patients admitted between March 2022 and October 2023 with a clinical diagnosis of sepsis according to the Sepsis-III definition [1] (patients with sepsis had a suspected infection and evidence of organ dysfunction with SOFA > 2). The patients with sepsis were further divided between patients with sepsis and patients with septic shock according to the septic shock diagnosis (lactate > 2 mmol/L and vasopressors were required to maintain the mean arterial pressure > 65 mmHg). The diagnosis of sepsis in the study group was based on the clinical diagnosis used in the participating ICUs, encoded in Margherita Prosafe, and centrally validated by the “Istituto Mario Negri” within the PROSAFE project [20]. The control group consisted of the population of critically ill patients admitted to the same ICU during the same period without sepsis diagnostic criteria. The exclusion criteria included age less than 18 years, active neoplasia, chronic myelomonocytic leukemia, chronic myeloid leukemia, chronic corticosteroid therapy (prednisone > 10 mg/day or equivalent), immunosuppressive or immunomodulatory therapy, and congenital immunodeficiency. For each patient, information was gathered concerning age, gender, nosological code of the clinical record, the originating hospital department, the date of admission to the ICU, the clinical diagnosis upon admission (for the division into “septic” versus “non-septic” study groups), the recent and remote pathological history (for the criteria of enrollment and exclusion from the study), the 28-day outcome, respective laboratory values (WBC, CRP, PCT, extended inflammatory parameters, creatinine), blood culture results, site of infection, and severity level at the time of admission using the SOFA score for the “septic” group. CRP, PCT, and creatinine were measured by using the automated chemical analyzer Beckman Coulter AU 5800 and the immunochemical analyzer Beckman Coulter Unicel DxI800 (Beckman Coulter, Brea, CA, USA) following the manufacturer’s recommendations. PCT was measured via chemiluminescence and expressed in ng/mL; CRP was measured through the turbidimetric method and expressed in mg/dL, and creatinine was measured through the immunoenzymatic assay and expressed in mg/dL. The focal point of our study was the extended inflammatory parameter NEUT-RI. To ascertain complete blood cell counts, a Sysmex XN hematology analyzer (Sysmex, Kobe, Japan) was employed, utilizing cytometry for the enumeration and classification of blood cells. To evaluate the effect of renal function on infection biomarkers, the patients were further divided into the subgroups “renal insufficiency” and “normal renal function”. The “renal insufficiency” subgroup included patients diagnosed with acute kidney injury (AKI) and chronic kidney disease (CKD) according to the Kidney Disease Improving Global Outcomes (KDIGO) classification [21]. The “sepsis” patients were further divided into the subgroups “uncomplicated sepsis” and “complicated sepsis” using the definition of septic shock (lactate > 2 mmol/L and the need for vasopressors to maintain MAP > 65 mmHg) [1]. The primary site of infection was recorded, creating seven subgroups for the following infection sites: lungs, abdomen, kidneys and urinary tract, skin and soft tissues, bloodstream, heart valves, and CNS. Finally, the 28-day outcome of the two groups of “septic” and “non-septic” patients was reported.

### Statistical Analysis

The data were collected in a database built and compiled using Microsoft Excel 365 in compliance with privacy regulations (Law 196/2003 and GDPR EU 679/2016). Statistical analyses of the data were conducted using the RSTUDIO 2023.12.1+402 statistical software. Since the Shapiro–Wilk normality test indicated that the NEUT-RI, WBC, PCT, and CRP variables were not normally distributed, a non-parametric statistical analysis was performed. The results were expressed as the median +/− interquartile range [IQR]. The independent samples *t*-test and non-parametric tests (Wilcoxon rank sum test) were used to investigate significant differences between groups and subgroups. The chi-squared test with Yates’ continuity correction for small samples was employed for the comparison of proportions. The Kruskal–Wallis rank sum test was conducted for the analysis of variance (for different infection sites). ROC analysis was performed to assess the performance of each biomarker and determine the best cut-off values. DeLong’s test for two correlated ROC curves was conducted for comparisons between ROC curves; the sensitivity and specificity tests for correlated ROC curves were conducted for comparison of sensitivity and specificity of the inflammatory parameters. A *p*-value ≤ 0.05 was considered statistically significant. Sensitivity and specificity for the diagnosis of sepsis upon admission to the intensive care unit of the NEUT-RI parameter were compared with those of PCT and CRP, along with their positive and negative predictive values. Between the two subgroups of “septic” patients (“uncomplicated sepsis” versus “complicated sepsis”), the values of each biomarker (NEUT-RI, PCT, and CRP) were compared. The “septic” and “non-septic” groups were further divided into subgroups of “renal insufficiency” and “normal renal function”, comparing the values of each biomarker. The trends of NEUT-RI and PCT were analyzed in the first 96 h of ICU stay in the “septic” group. The “septic” and “non-septic” groups were further divided into sub-groups of “alive” versus “deceased” based on the 28-day outcome, and the values of each biomarker were compared. Sensitivity and specificity for predicting the 28-day outcome upon admission to the ICU of the NEUT-RI parameter were also compared with those of PCT and CRP, along with their positive and negative predictive values.

## 3. Results

A total of 200 patients were included in this study (Figure 1). The average age was 73 years (59–79) with males representing 50.5% (*n* = 101) and females representing 49.5% (*n* = 99). In the “septic” group, 89 patients (44.5%) were included vs. 111 patients (55.5%) in the “non-septic group” (Table 1, Panel A). The admission diagnosis of the “septic” patients was pneumonia or upper respiratory tract infection in 39 patients (42.8%); secondary peritonitis due to cholangitis, intestinal obstruction, or intestinal perforation in 25 patients (27.5%); urinary tract infection or pyelonephritis in 17 patients (18.7%); infection of the skin and soft tissues, including necrotizing fasciitis, gas gangrene, infections of orthopedic prostheses, or submandibular abscesses, in 7 patients (7.7%); heart valve infections in 2 patients (2.2%); and bloodstream infection in 1 patient (1.1%). No patients in our population had CNS infections. Among the “septic” patients, 49 (53.8%) belonged to the “uncomplicated sepsis” subgroup, and 42 (46.2%) had a diagnosis of septic shock upon admission to the ICU (the “complicated sepsis” subgroup) (Table 1, panel B). The admission diagnosis upon entry into the ICU for the “non-septic” patients was as follows: post-operative monitoring in cardiac patients or those with severe OSAS on nocturnal CPAP (44 (39.6%)); post-anoxic coma (18 (16.2%)); inappropriate substance intake (12 (10.9%)); acute respiratory failure (11 (10%)), including patients with motor neuron disease onset, acute pulmonary edema, pleural effusion, pulmonary embolism, or h exacerbation of COPD; post-cardiovascular events (9 (8.1%)); hemorrhagic shock (4 (3.6%)); electrolyte imbalance or metabolic acidosis (4 (3.6%)); post-neurological events, including cerebral stroke, intracranial hemorrhage, or status epilepticus (5 (4.5%)); neuroleptic syndrome (2 (1.8%)); anaphylactic shock (1 (0.9%)); and heat stroke (1 (0.9%)).

### 3.1. Inflammatory Parameters

A significant difference between the “septic” and “non-septic” groups was detected in the NEUT-RI plasma concentration (53.80 [49.65–59.05] vs. 48.00 [46.00–49.90], *p* < 0.001, respectively) (Figure 1, panel A). Similar results were detected for PCT and CRP (PCT 8.83 [0.82–45.88] vs. 0.48 [0.29–1.64], *p* < 0.001; CRP 20.79 [12.54–118.91] vs. 6.68 [1.54–21.70], *p* < 0.001). The NEUT-RI and PCT were able to discriminate between not complicated sepsis and septic shock (PCT 1.71 [0.42–12.09] vs. 32.59 [8.83–100.00], *p* < 0.001 and NEUT-RI 51.50 [47.80–56.30] vs. 56.20 [52.30–61.92], *p* = 0.005). There were no differences in NEUT-RI and CRP depending on the site of infection in the “septic” group. Regarding the site of infection, PCT was higher in patients with kidney or urinary tract infection compared to lung infection (88.50 [24.20–113.00] vs. 1.64 [0.39–12.54], *p* < 0.001) (Appendix A).

### 3.2. Renal Failure and Inflammatory Parameters

The number of patients affected by renal failure upon admission to the ICU was 70 (35%), including 47 AKI and 23 CKD. The inflammatory parameters are described in the Appendix A. The comparison between patients based upon the presence or the absence of renal insufficiency is described in Table 2. In the “septic” group, 43 patients (48.3%) had renal failure upon admission, specifically 36 with AKI (including 5 with AKI in CKD) and 7 with CKD. In the “non-septic” group, 27 patients (24.3%) had renal failure upon admission, specifically 11 with AKI (including 2 with AKI in CKD) and 16 with CKD. NEUT-RI, PCT, and CRP values were significantly different in the patients with “renal failure” than those with “normal renal function” in the overall population (Appendix A). Within the septic group, no statistically significant difference was found in NEUT-RI and CRP (55.10 [52.15–59.05] vs. 51.70 [47.82–58.65]; 25.53 [17.82–148.22] vs. 18.80 [7.35–103.27], respectively), while a statistically significant difference was found in PCT (32.23 [5.86–83.72] vs. 1.79 [0.39–13.04], *p* < 0.001). In the “non-septic” group, only CRP exhibited a significant difference between patients with or without renal function impairment (13.45 [6.03–130.85] vs. 3.78 [1.05–18.59], *p* = 0.003). Moreover, NEUT-RI and PCT were stratified into severity classes according to SAPS II; both inflammatory parameters showed a positive correlation (*p* < 0.001, linear regression) with SAPS II but with a great dispersion of data (Appendix A).

Both NEUT-RI and PCT showed a statistically significant difference when measured at different timepoints in the “septic” group (*p* < 0.01 repeated measures ANOVA for both inflammatory parameters). In particular, both NEUT-RI and PCT were lower at 96 h compared to the admission time and 48 h. Neither NEUT-RI nor PCT were lower at 48 h compared to the admission time (Appendix A). The performance of each inflammatory parameter for the diagnosis of sepsis and their best cut-off values are described in Appendix A, while the comparisons in AUROC are shown in Figure 2. There were no statistically significant differences between the AUROC of the three parameters (NEUT-RI vs. PCT *p* = 0.83, NEUT-RI vs. CRP *p* = 0.29, DeLong’s test for two correlated ROC curves). The CRP specificity was statistically different from the NEUT-RI and the PCT specificity (*p* = 0.01 and *p* = 0.007, respectively, specificity test for two correlated ROC curves). Regarding the 28-day outcome, the overall mortality in our population was 17.5%; there was a trend in higher mortality in the “septic” compared to the “non-septic” groups (23.1% vs. 12.8%, *p* = 0.09).

The figure shows the area under the receiver operating characteristic (ROC) curve for the distinction of inflammatory parameters for detection of sepsis. The areas under the ROC curves are as follows: NEUT-RI (continuous line): 0.79 [95% CI 0.74–0.91]; PCT (dashed line): 0.76 [95% CI 0.77–0.93]; CRP (dotted line): 0.73 [95% CI 0.77–0.93; *p* < 0.001. NEUT-RI = neutrophil-reactive intensity; PCT = procalcitonin; CRP = C-reactive protein.

### 3.3. 28-Day Outcomes

The comparison of inflammatory parameters for 28-day outcomes between “survivors” and “deceased” in the “septic” and “non-septic” patient groups is described in Table 3. We excluded from the analysis 18 patients who were admitted to the ICU after cardiac arrest, where a poor prognosis was based on neurological data. NEUT-RI and PCT (not CRP) at admission in the ICU in the septic group were higher in patients who died compared to patients who survived (58.80 [53.85–73.10] vs. 53.05 [48.90–57.22], *p* = 0.005 and 39.56 [17.39–83.72] vs. 3.22 [0.59–32.32], *p* = 0.002, respectively). No differences were found in the “non-septic” group. A comparison of NEUT-RI and PCT values for discriminating between “alive” and “deceased” at 28 days is shown in Appendix A. There were no statistically significant differences between AUROC, sensitivity, and specificity of the two inflammatory parameters. Both NEUT-RI and PCT showed a high negative predictive value (NPV) (93.0 [87.17–96.76] and 95.10 [88.93–98.39], respectively) and low positive predictive value (PPV) (28.30 [16.8–42.34] and 38.09 [23.6–54.3], respectively).

## 4. Discussion

The main findings of this new retrospective analysis are as follows: (1) NEUT-RI, PCT, and CRP levels were significantly elevated in critically ill patients diagnosed with sepsis compared to those admitted for other causes; (2) Both NEUT-RI and PCT exhibited high accuracy in sepsis diagnosis, showing an early increase shortly after the onset of systemic inflammation, which could redefine their value as crucial parameters for the early diagnosis of sepsis; (3) NEUT-RI emerged as a more effective biomarker in cases of renal failure; (4) Both NEUT-RI and PCT appeared to predict 28-day mortality upon admission to the ICU.

Sepsis, as commonly understood, triggers immune system activation, leading to changes in inflammatory markers [22]. New extended inflammatory parameters like NEUT-RI, which are closely related to the appearance of immature granulocytes and neutrophil activation, are now available with just a complete blood count [16]. Our study corroborates this phenomenon, revealing a significant difference between the “septic” and “non-septic” patient groups in terms of PCT, CRP, and the extended inflammatory parameter NEUT-RI. Through the inclusion of a larger population compared to that of the previous study [19], NEUT-RI has once again demonstrated worthy diagnostic efficacy. The optimal cut-off value is identified as 50.75 FI, surpassing the previous threshold of 51.9 FI. Although sensitivity has decreased slightly to 70.9% from the prior 80.4%, there is a notable improvement in specificity, now standing at 80.7% as opposed to the earlier 76%.

In our investigation, PCT emerged as a valuable biomarker for sepsis diagnosis, maintaining a consistent best cut-off value of 2.17 ng/mL, in line with the previous study [19]. The sensitivity, though slightly reduced to 62.9% from the earlier 69.6%, is counterbalanced by a worthily high specificity of 82.9%, only marginally lower than that of the previous study [19]. CRP, acknowledged as an acute-phase inflammatory marker indicative of the acute phase of sepsis [23], confirmed its diagnostic utility in our study. The results demonstrated a robust sensitivity of 84.6%, surpassing the prior 80.4% [19]. However, this gain in sensitivity is accompanied by a decrease in specificity to 56.2% compared to the earlier 70.7%. This observation is certainly not new and is consistent with several previously described findings [24]. Regarding the positive predictive value, both NEUT-RI and PCT exhibited similar results. However, NEUT-RI showcased a superior negative predictive value of 77.2% compared to 63.7% for PCT. This reaffirms the potential diagnostic efficacy of NEUT-RI in minimizing the risk of false negatives and enhancing the likelihood of accurately identifying individuals with sepsis [18,25]. Concerning the discriminatory capacity of various inflammatory biomarkers in assessing the severity of ongoing sepsis, our study, benefiting from an increased sample size, uncovered novel and noteworthy evidence. The extended inflammatory parameter NEUT-RI could help clinicians recognize early-stage sepsis with an uncomplicated course from sepsis that will evolve into septic shock. PCT further demonstrated its utility as a biomarker in this context, revealing a notable distinction between the two examined subgroups “not complicated sepsis” and “septic shock.” This insight holds potential for guiding the initial management of septic patients, facilitating more precise and timely therapeutic interventions, and aiding in the prompt identification of complications. We selected renal function as a parameter for sub-analyses due to its known substantial impact on sepsis biomarkers [26]. CRP is acknowledged to have a negative correlation with glomerular filtration rate, yet it remains a reliable predictor of infection in individuals with compromised renal function [27,28]. PCT, being eliminated through renal clearance, may yield elevated values in patients with acute renal failure, potentially leading to increased levels even in the absence of infections [29,30]. Consequently, its sensitivity for bacterial infection diagnosis could be compromised, particularly considering the unknown optimal cut-off value in acute renal failure cases and the unclear relationship between creatinine/urea and PCT values. Our analysis supported the idea that PCT in septic patients is indeed influenced by renal function, challenging its reliability as a predictive biomarker for sepsis. Conversely, as previously highlighted [15,18,19], NEUT-RI appears to be a more reliable biomarker in these scenarios. NEUT-RI exhibited a significant difference between the “renal failure” and “normal renal function” groups, yet this distinction was not corroborated within the individual “septic” and “non-septic” groups. This suggests that NEUT-RI is less susceptible to the influence of renal function, and the initially observed difference may be attributed to the disparate incidence of renal insufficiency between the “septic” and “non-septic” groups with markedly higher creatinine values in the “septic” group. In our investigations, CRP also emerged as a valuable contributor to sepsis diagnosis, as it did not show a discernible influence by renal function in septic patients. Finally, the analysis of the 28-day outcome revealed that, in septic patients, NEUT-RI exhibited an AUC greater than 0.7 with a sensitivity of 62.5% and specificity of 76.9% at a best cut-off value of 53.6 FI. In comparison, procalcitonin demonstrated an AUC of 0.76 with a sensitivity of 76.2% and specificity of 79.9% at a best cut-off value of 12.18 ng/mL. No statistically significant differences were found in AUC, sensitivity, and specificity. Both NEUT-RI and PCT exhibited a low positive predictive value (28.30 and 38.09, respectively) but a high negative predictive value (93% and 95.1%).

This study has several noteworthy limitations that warrant acknowledgment. First, the comparison of NEUT-RI values with those of PCT and CRP in the early stages of sepsis onset, particularly in the emergency department or ward, was not consistently feasible. This discrepancy arises from the fact that these values were not universally requested at the initial phase but were only analyzed upon admission to the ICU. Furthermore, in five septic patients admitted, PCT levels exceeded 250 ng/mL, and in two patients, it surpassed 100 ng/mL. As a result, an approximation of these values to 250 ng/mL and 100 ng/mL, respectively, was necessary to facilitate statistical analyses. In exploring the potential correlation between NEUT-RI and PCT with the 28-day mortality of septic patients admitted to the ICU, stratification into severity classes (such as with SAPS II) could not be statistically significant due to the small sample size. This limitation underscores the need for further investigations in the future. Additionally, it would be valuable to correlate the insights derived from NEUT-RI with IL-6 levels for early sepsis diagnosis. As indicated in the literature, IL-6 plays a pivotal role in determining the prognosis of patients in the ICU [31]. Integrating IL-6 data with the previously obtained information on NEUT-RI and PCT has the potential to enhance the specificity and/or sensitivity of the outcome, offering a more comprehensive understanding. Thus, to summarize the disadvantages of using NEUT-RI in common clinical practice, we can state the following: Coexisting pharmacological (for example steroids or immunosuppressive therapy) or pathological causes of neutrophilia or neutropenia other than sepsis could lead to misinterpretation of patient conditions; cytometry is used for the enumeration and classification of blood cells, so adequate instrumentation is essential to obtain this data; physicians are actually not confident with the NEUT-RI parameter of sepsis evaluation, in particular with the high negative predictive value of this parameter, which helps the clinician to exclude, not to confirm, sepsis.

## 5. Conclusions

The inflammatory biomarkers examined in this study have demonstrated their efficacy in aiding physicians during the initial phase of sepsis diagnosis. Specifically, the elevation of NEUT-RI values appears to be a valuable asset in early sepsis detection, effectively distinguishing the severity of sepsis upon admission. Furthermore, it could aid in identifying septic patients at a higher risk of progressing to septic shock with unfavorable long-term outcomes. Nevertheless, further integration of these findings into additional studies is necessary to confirm their reliability and applicability across broader contexts.

## Figures and Tables

**Figure 1 diagnostics-14-00821-f001:**
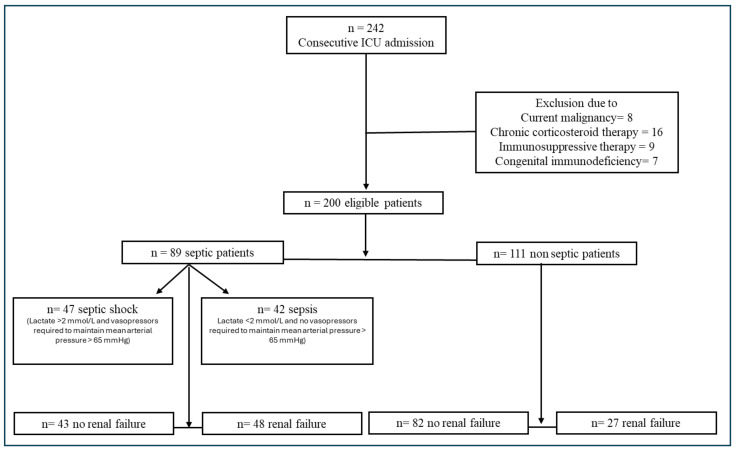
Study design flow chart.

**Figure 2 diagnostics-14-00821-f002:**
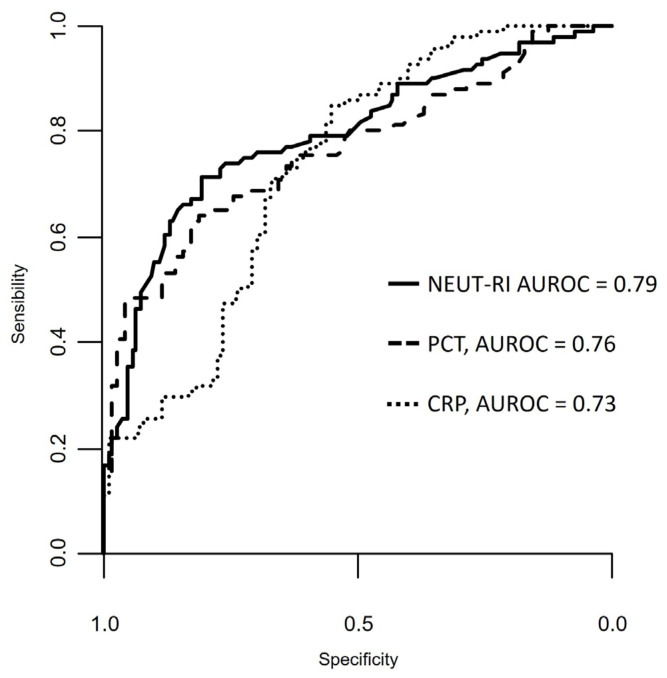
Comparison of ROC curves for NEUT-RI (FI), PCT (ng/mL), and CRP (mg/dL) for sepsis; AUC of NEUT-RI is higher than that of PCT and CRP.

**Table 1 diagnostics-14-00821-t001:** Comparisons of patients’ characteristics and inflammatory parameters divided into “septic” and “non-septic” patients (Panel A) and, among septic, into “not complicated sepsis” or “septic shock” patients (Panel B).

	A			B		
	Septic (*n* = 89)	Non-Septic (*n* = 111)	*p*	Not Complicated Sepsis (*n* = 47)	Septic Shock (*n* = 42)	*p*
Age	73.00 [63.00–79.00]	73.00 [56.00–79.00]	0.454	74.00 [64.00–78.00]	73.00 [62.75–79.75]	0.808
Male%	48.4	52.3	0.679	59.6	35.7	0.042
Creatinine	1.57 [0.85–2.85]	1.00 [0.77–1.39]	<0.001	1.16 [0.77–1.64]	2.18 [1.67–3.67]	<0.001
CRP	20.79 [12.54–118.91]	6.68 [1.54–21.70]	<0.001	18.79 [7.78–149.16]	23.03 [17.97–87.09]	0.7244
PCT	8.83 [0.82–45.88]	0.48 [0.29–1.64]	<0.001	1.63 [0.40–12.09]	32.59 [8.83–100.00]	<0.001
NEUT-RI	53.80 [49.65–59.05]	48.00 [46.00–49.90]	<0.001	51.5 [47.80–56.30]	56.20 [52.30–61.92]	0.0054

**Table 2 diagnostics-14-00821-t002:** Comparison of inflammatory parameters among “septic” and “non-septic” patients based on renal function.

	Septic			Non-Septic		
	Renal Failure (*n* = 43)	Normal Renal Function (*n* = 48)	*p*	Renal Failure (*n* = 27)	Normal Renal Function (*n* = 82)	*p*
Age	73.00 [63.00–79.00]	72.50 [63.50–77.50]	0.443	78.00 [73.00–81.00]	68.00 [53.25–77.00]	0.004
Male%	37.2	58.3	0.071	51.9	52.4	1
Creatinine	2.81 [1.83–4.26]	0.86 [0.74–1.28]	<0.001	1.60 [1.33–2.59]	0.89 [0.68–1.19]	<0.001
CRP	25.53 [17.82–148.22]	18.80 [7.35–103.27]	0.164	13.45 [6.03–130.85]	3.78 [1.05–18.59]	0.003
PCT	32.23 [5.86–83.72]	1.79 [0.39–13.04]	<0.001	1.15 [0.31–3.63]	0.47 [0.28–1.21]	0.193
NEUT-RI	55.10 [52.15–59.05]	51.70 [47.82–58.65]	0.101	47.90 [46.40–50.05]	48.00 [45.85–49.88]	0.886

**Table 3 diagnostics-14-00821-t003:** Comparison of inflammatory parameters for 28-day outcomes between “survivors” and “deceased” in the “septic” and “non-septic” patient groups.

	Septic			Non-Septic		
	Alive (*n* = 78)	Dead (*n* = 11)	*p*	Alive (*n* = 95)	Dead (*n* = 16)	*p*
CRP	21.52 [11.48–137.89]	19.06 [14.10–36.49]	0.772	7.28 [1.88–21.70]	1.06 [0.41–80.00]	0.477
PCT	3.22 [0.59–32.32]	39.56 [17.39–83.72]	0.002	0.46 [0.28–1.21]	0.72 [0.57–1.10]	0.453
NEUT-RI	53.05 [48.90–57.22]	58.80 [54.45–73.35]	0.005	47.90 [45.80–49.82]	45.60 [44.00–47.60]	0.184

## Data Availability

Data is unavailable due to privacy.

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
