# Peer review of "A Secondary Retrospective Analysis of the Predictive Value of Neutrophil-Reactive Intensity (NEUT-RI) in Septic and Non-Septic Patients in Intensive Care"

_diagnostics, 2024, doi:10.3390/diagnostics14080821_

Round 1

Reviewer 1 Report

Comments and Suggestions for Authors

Dear Authors,

Thank you for your manuscript. After a thorough reading I summarize my concerns and question regarding the paper.

line 47 Currently, blood cultures are considered the gold standard for pathogen isolation and sepsis diagnosis [6]. – I challenge this sentence in 2024 with the advent of molecular genetic diagnosis. I never waited for the result of blood culture to make a diagnosis of sepsis.

line 72 PCR, PCT, extended inflammatory parameters, creatinine) – did you mean CRP?

line 74 The infection site and the 28-day outcome were added to this existing data – why was the infection site needed?

line 83 The control group consisted of the population of critically ill patients admitted to the same ICU during the same period without sepsis diagnostic criteria – I feel this results in a significant inhomogeneity, could you please specify what other diseases were the admission criteria?

line 89 what do you mean by nosologically code?

line 91 what is the recent and remote pathological history? Why is it relevant?

line 167 There were no differences in NEUT-RI and CPR – again, did you mean CRP? – see line 184, 202, Figure 2, Table 1, Table 2 and Table 3 again for the same misspelling.

line 271 CRP, acknowledged as an acute-phase inflammatory marker indicative of the acute phase of sepsis [24], confirmed its diagnostic utility in our study – the cited literature is an analysis of use of CRP in the emergency department in different infections, I do not think that this can be the base of stating that CRP is useful in the diagnosis of sepsis. It is not.

Remarks

1.       1. I find the use of PCT and CRP in renal failure is not the best idea. The reasons for this are given below.

Renal failure can lead to elevated PCT levels, as the kidneys are a primary site for PCT clearance. In patients with renal insufficiency, the reduced ability to clear PCT from the bloodstream can result in higher baseline levels, independent of infection. The elevated baseline levels of PCT in renal failure patients make it more challenging to interpret increases solely as indicators of bacterial infection. Clinicians must consider renal function when evaluating PCT levels and may need to rely on other clinical signs, symptoms, and diagnostic tests to accurately diagnose infections.

The impact of renal failure on CRP levels is less direct than with PCT. CRP is primarily synthesized in the liver in response to inflammation, and its levels can be affected by a variety of factors, including the presence of chronic diseases and infections. Patients with renal failure, particularly those on hemodialysis, often have elevated CRP levels due to the chronic inflammatory state associated with renal insufficiency. This elevation can be attributed to the body's response to the disease itself, as well as to other factors like dialysis-related complications.

2.       2. Despite referring to blood cultures as gold standard for sepsis diagnosis, there are no indicators that any microbiological results were involved in the study. It is more interesting considering that PCT is a biomarker of bacterial infection with levels staying low in viral and fungal sepsis. There is no indication what was the percentage of bacterial sepsis in the studied population?

 Best regards,

Dr. Peter L. Kanizsai

Comments on the Quality of English Language

This paper is of good quality of English, however misspelling of CRP in a disturbing number must be corrected.

Reviewer 2 Report

Comments and Suggestions for Authors

Dear Editor

In the current manuscript, the authors aimed to assess the predictive value of

 Neutrophil-reactive intensity (NEUT-RI) in diagnosing sepsis and to evaluate its prognostic significance in distinguishing 28-day mortality outcomes via using a secondary retrospective observational analysis. Results clarified a significant difference between the "septic" and "non-septic" groups in the NEUT-RI plasma concentration. They concluded that the inflammatory biomarkers assessed in this study offer valuable support in the early diagnosis of sepsis. NEUT-RI elevation appears particularly promising for early sepsis detection and severity discrimination upon admission.  However,

·       The aim of the study is unclear and needs rephrasing is it predictive or diagnostic value, prognostic significance is enough no need to be in distinguishing mortality outcome.

·       What is the measuring unit for Neutrophil-reactive intensity (NEUT-RI)???

·       Is the study for Diagnostic or prognostic significance?

·       Some abbreviations are mentioned without the complete name for the 1st time.

·       Inclusion & exclusion criteria are not clear.

·       Diagnostic criteria for sepsis are not mentioned (regarding who will be included in the study)

·       Study design is not clear

·       Statistical analysis is very long

·       Other comments in the manuscript file.

·       Conclusion is vague and how many markers are studied so better to be rephrased

  Best regards

Comments on the Quality of English Language

Good

Reviewer 3 Report

Comments and Suggestions for Authors

The authors described "A secondary retrospective analysis of the predictive value of Neutrophil-Reactive Intensity (NEUT-RI) in Septic and Non-septic Patients in Intensive Care".

As they mentioned, NEUT-RI elevation may be useful for early sepsis detection and severity discrimination upon admission.

In this manuscript, both NEUT-RI and PCT exhibited high accuracy in sepsis diagnosis, showing an early increase shortly after the onset of systemic inflammation. This could redefine their value as crucial parameters for the early diagnosis of sepsis. I have some suggestions to improve this manuscript.

1. First of all, there are a lot of typos (e.g. CPR in Figure 2, Table 1, Discussion,,,, or cased ? in line 277). Please read the paper carefully again to make sure there are no mistakes.

2. The advantages of NEUT-RI compared to PCT were well documented, but what about the disadvantages?

3. They described "To evaluate the effect of renal function on infection biomarkers, patients were further divided into subgroups "renal insufficiency" and "normal renal function" in Method. Why did you decide to evaluate the renal function? Please add it in Introduction.

Comments on the Quality of English Language

There are a lot of typos.

Round 2

Reviewer 1 Report

Comments and Suggestions for Authors

Dear Authors,

Thank you for the reapid response and clarification.

Best regards,

Dr. P. Kanizsai

Reviewer 2 Report

Comments and Suggestions for Authors

Dear all

The corrections are convenient however, the data in the supplementary files are not updated as the main manuscript. Also, the number of patients enrolled in the study is better to be mentioned in the methods not the results of the abstract.

Best Regards

Reviewer 3 Report

Comments and Suggestions for Authors

The authors revised the manuscript precisely. However, there are still typos (e.g. CPR in Figure2 that I already pointed out). I recommend the authors reread the manuscript in details.

Comments on the Quality of English Language

none
